# Letter to the Editor: SFlt-1 and PlGF Levels in Pregnancies Complicated by SARS-CoV-2 Infection

**DOI:** 10.3390/v13122377

**Published:** 2021-11-27

**Authors:** Valentina Giardini, Sara Ornaghi, Eleonora Acampora, Maria Viola Vasarri, Francesca Arienti, Carlo Gambacorti-Passerini, Marco Casati, Andrea Carrer, Patrizia Vergani

**Affiliations:** 1Department of Obstetrics and Gynecology, MBBM Foundation, San Gerardo Hospital, University of Milano-Bicocca, 20900 Monza, Italy; sara.ornaghi@gmail.com (S.O.); e.acampora@campus.unimib.it (E.A.); m.vasarri@campus.unimib.it (M.V.V.); f.arienti9@campus.unimib.it (F.A.); patrizia.vergani@unimib.it (P.V.); 2Hematology Division, ASST-Monza, San Gerardo Hospital, University of Milano-Bicocca, 20900 Monza, Italy; carlo.gambacorti@unimib.it (C.G.-P.); mail.carrer@gmail.com (A.C.); 3Laboratory Medicine, ASST-Monza, San Gerardo Hospital, University of Milano-Bicocca, 20900 Monza, Italy; m.casati@asst-monza.it

**Keywords:** COVID-19, SARS-CoV-2, angiogenic factors, sFlt-1, endothelial dysfunction

## Dear Editor 

We read with interest the work by Espino-y-Sosa and colleagues [1], who recently reported that the Soluble Fms-like Tyrosine Kinase-1/Angiotensin-II (sFlt-1/ANG-II) ratio could be a good predictor of adverse outcomes, including pneumonia, intensive care unit (ICU) admission, intubation, viral sepsis, and death, among pregnant women with COVID-19.

COVID-19 is a respiratory infection characterized by signs and symptoms associated with the dysfunction of the renin–angiotensin system (RAS). RAS activation leads to the formation of ANG II, a powerful vasoconstrictor and promotor of inflammation, fibrosis, and coagulation [2]; it also induces the release of sFlt-1 in case of hypoxia [3]. sFlt-1 is the soluble receptor of placental growth factor (PlGF), a potent angiogenic factor, and it antagonizes PlGF’s activity in the circulation, thus creating an anti-angiogenic state and ensuing endothelial dysfunction (ED) [4]. 

SARS-CoV-2 invades the respiratory mucosa of the host via the Angiotensin-Converting Enzyme 2 (ACE2) receptor. ACE2 is an important element of RAS [5], and catalyzes ANG II into angiotensin 1–7 (ANG 1–7), a vasodilator with the opposite function of ANG II. Use of the ACE2 receptor for viral entry leads to the downregulation of its synthesis, with the subsequent amplification of ANG II actions and, in turn, vasospasm, inflammation, microvascular thrombosis, and organ damage. In line with these observations, a linear association between serum ANG II levels and viral load and lung damage in COVID-19 patients has been reported [6].

Preeclampsia (PE), a pregnancy-specific hypertensive disorder with multisystem involvement, is characterized by increased sensitivity to ANG II and ED, with high sFlt-1 and low PlGF values. Currently, the sFlt1/PlGF ratio is used as a clinical biomarker for the early detection and prognosis of PE [7].

Considering the common RAS-mediated underlying etiopathogenesis, we initially investigated the sFlt1/PlGF ratio in non-pregnant patients with COVID-19 pneumonia. We originally identified the presence of an angiogenic imbalance in COVID-19 patients, similar to that identified in PE women [8]. Precisely, levels of sFlt-1 were significantly higher in COVID-19-related pneumonia cases compared to those with pneumonia due to other causes and to healthy controls. PlGF values were not significantly affected by COVID-19, but the sFlt1/PlGF ratio was substantially higher in COVID-19-positive compared with COVID-19-negative pneumonia. Subsequently, other authors confirmed the increased sFlt-1 values in severe COVID-19 and identified sFlt-1 as a biomarker to predict survival and thrombotic accidents in COVID-19 patients [9].

The role of angiogenic markers in pregnancies complicated by SARS-CoV-2 infection is still unclear, possibly because of the interference of the placenta, the major extrarenal RAS site during pregnancy [10].

We then proceeded to assess the sFlt1/PlGF ratio in pregnant women with SARS-CoV-2 infection. Assessment of this ratio could be helpful in guiding the management of these patients and improving the understanding of the pathophysiology of SARS-CoV-2 infection in pregnancy. In fact, an increased incidence of PE among COVID-19 mothers has been reported, although such association is still incompletely elucidated [11,12]. In addition, a study showed that a PE-like syndrome can be induced by severe COVID-19 during pregnancy [13]. 

Precisely, we conducted a retrospective analysis of positive SARS-CoV-2 pregnant women admitted to our center from April 2020 to October 2021. SARS-CoV-2 infection was diagnosed by RT-PCR assay on nasopharyngeal swabs. Serum dosage of sFlt-1 and PlGF (Cobas e801 analyzer Roche platform, Roche Diagnostics) was performed at the time of diagnosis of SARS-CoV-2 infection, before the beginning of any therapy. Patients already on therapy (enoxaparin sodium, steroids, or hydroxychloroquine) as well as patients without a chest X-ray performed at the time of hospital admission were excluded. The study population included 57 pregnant women, who were divided into two groups (Table 1): women with signs and symptoms of COVID-19 at hospital presentation (*n* = 20, 35%) and asymptomatic women (*n* = 37, 65%). sFlt-1/PlGF ratio was stratified using cut-off values clinically utilized for PE prediction (low risk < 38, high risk > 85 if before 34 weeks’ gestation or > 110 if after 34 weeks’ gestation) [14].

Asymptomatic women were identified at a surveillance swab required before hospital admission for obstetric reasons in 86% of cases (*n* = 32). The mean gestational age at diagnosis of SARS-CoV-2 infection was 37^5/7^ weeks in asymptomatic women and 32^0/7^ weeks in symptomatic women (*p* = 0.089). Of note, 4 (11%) asymptomatic women had radiological evidence of pneumonia compared to 16 (80%) in the symptomatic group. In Espino-y-Sosa’s work [1], none of the pregnant women with non-severe COVID-19 had pneumonia; it is not clear whether patients underwent radiological examination upon hospital admission. 

Among our 20 COVID-19 symptomatic cases, 7 (35%) required high-dependency/intensive care with 3 of them undergoing endotracheal intubation. Continuous positive airway pressure was applied in five cases (25%). Eight (40%) women delivered by cesarean section but only in two cases (10%) for respiratory failure. The mean gestational age at delivery was 38^2/7^ weeks with a mean latency time between symptom onset and delivery of 48 ± 44 days. All women received therapy with enoxaparin sodium, 14 (70%) were given steroids, and 2 (10%) hydroxychloroquine (first pandemic wave). Of note, there were no stillbirths, or maternal or neonatal deaths, among these symptomatic patients; additionally, no cases of PE, fetal growth restriction, or small for gestational age neonates were diagnosed. Espino-y-Sosa et al. reported worse maternal and neonatal outcomes compared with our results.

Additionally, differently from Espino-y-Sosa et al., we identified higher sFlt-1 levels in asymptomatic patients compared to symptomatic patients (4899 ± 4357 vs. 3187 ± 2426 pg/mL, *p* = 0.005). sFlt-1/PlGF ratio at admission was ≤38 in 18 of the 20 symptomatic women compared to 22 (59%) of the asymptomatic patients (mean gestational age at admission 32^1/7^ weeks versus 38^3/7^ weeks) (*p* = 0.018). In turn, rates of patients with sFlt-1/PlGF ratio at admission > 85/110 were similar between symptomatic and asymptomatic group (*n* = 0 versus *n* = 4, 11%; *p* = 0.286).

This difference in the increase in sFlt-1 between our data and those reported by Espino-y-Sosa and colleagues may be due to the higher gestational age at hospital admission of our asymptomatic patients. An additional explanation might be the delayed hospital access of Espino-y-Sosa’s severe COVID-19 patients, leading to increased sFlt-1 levels due to prolonged exposure to hypoxia [3]. In our study, the mean number of days between symptom onset and hospital admission was five. Furthermore, we performed the assay of sFlt-1 and PlGF on serum according to the instructions of Roche Diagnostics; in Espino-y-Sosa’s work, it is unclear whether sFlt-1 and PlGF tests were performed on plasma or serum and whether these women received pre-hospital treatment that may have affected the markers analyzed.

No significant differences in the incidence of PE were identified between severe and non-severe COVID-19 patients in Espino-y-Sosa’s work, as well as in our cohort. A recent sub-analysis from the INTERCOVID study population showed that COVID-19 during pregnancy is independently associated with PE. Interestingly, this association is not modified by COVID-19 severity [15]. sFlt-1/PlGF ratio results among SARS-CoV-2-infected asymptomatic women are of particular interest. If the increase in sFlt-1 levels we observed is the consequence of viral infection, and not just of the higher gestational age at its evaluation, then the identification of asymptomatically infected pregnant women might be important since they may benefit from more intensive antenatal surveillance of fetal growth and blood pressure due to a potential increased risk of developing PE. Furthermore, the data from this study provide further reason to push pregnant women to be vaccinated, given the possible obstetric complications in the case of asymptomatic SARS-CoV-2 infection. In conclusion, our data suggest that SARS-CoV-2 infection during pregnancy could influence the angiogenic profile, but likely with a different effect according to the type of viral infection (symptomatic versus asymptomatic). Further research in a larger prospective cohort is needed.

## Figures and Tables

**Table 1 viruses-13-02377-t001:** Population characteristics comparing positive SARS-CoV-2 pregnant women with signs and symptoms of COVID-19 at admission to asymptomatic women.

SARS-CoV-2 Pregnant Women	All	Asymptomatic	Symptomatic	*p* Value
Variables	*n* = 57	37 (65)	20 (35)	
**Anamnestic Characteristics**				
Age (years)	33 ± 5	33 ± 5	33 ± 4	0.667
Italian	30 (53)	20 (54)	10 (50)	0.788
Nulliparous	23 (40)	14 (38)	6 (30)	0.772
Obesity (BMI > 30 kg/m^2^)	12 (21)	7 (19)	5 (25)	0.736
Diabetes/Gestational diabetes mellitus	15 (26)	10 (27)	5 (25)	0.580
Chronic hypertension	2 (4)	2 (5)	0	0.536
**SARS-CoV-2 infection**				
GA at positive swab (weeks.days ± weeks)	35.5 ± 6	37.5 ± 5	32 ± 6	0.089
Respiratory symptoms at admission	16 (28)	0	16 (80)	0.0001
Pneumonia on chest x-ray	20 (35)	4 (11)	16 (80)	0.0001
High dependency unit admission	6 (11)	0	6 (30)	0.001
ICU admission	3 (5)	0	3 (15)	0.039
Oxygen supplementation	14 (25)	0	14 (70)	0.0001
Continuous positive airway pressure (CPAP)	5 (9)	0	5 (25)	0.004
Intubation	3 (5)	0	3 (15)	0.039
Maternal/fetal/neonatal death	0	0	0	NA
**Angiogenic factors**				
GA at blood test (weeks.days ± weeks)	36.2 ± 6	38.3 ± 4	32.1 ± 6	0.208
Latency timeCOVID-19 symptoms—blood test (days)	14 ± 24	27 ± 34	5 ± 4	0.038
sFlt-1 (pg/mL)	4948 ± 3988	4899 ± 4357	3187 ± 2426	0.005
PlGF (pg/mL)	237 ± 178	178 ± 104	346 ± 232	0.099
sFlt1/PlGF ratio	38 ± 50	50 ± 58	17 ± 23	0.099
sFlt1/PlGF < 38	40 (70)	22 (59)	18 (90)	0.018
sFlt1/PlGF > 85/110 (if before/after 34 weeks)	4 (7)	4 (11)	0	0.286
**Pregnancy**				
Hypertensive disorders of pregnancy/post-partum	5 (9)	4 (11)	1 (5)	0.647
Fetal growth restriction	3 (5)	3 (8)	0	0.545
Premature birth < 37 weeks	4 (7)	0	4 (20)	0.012
Small for gestational age newborn	6 (11)	6 (16)	0	0.081

Data presented as mean ± standard deviation or *n* (%). GA: gestational age. Fetal growth restriction: Delphi consensus methodology [16]. Small for gestational age newborn: birthweight below the 10th percentile (INeS charts) [17].

## Data Availability

The data that support the findings of this study are available on request from the corresponding author.

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
