# Peer review of "Letter to the Editor: SFlt-1 and PlGF Levels in Pregnancies Complicated by SARS-CoV-2 Infection"

_viruses, 2021, doi:10.3390/v13122377_

Round 1
Reviewer 1 Report
In this letter to the editor, the authors respond to an article by Espino-y-Sosa and colleagues (Espino-y-Sosa, S, et al. Novel Ratio Soluble Fms-like Tyrosine Kinase-1/Angiotensin-II (sFlt-1/ANG-II) in Pregnant Women Is Associated with Critical Illness in COVID-19. Viruses 2021, 13, 1906. https://doi.org/10.3390/v13101906), who recently reported that the Soluble Fms-like Tyrosine Kinase-1/Angiotensin-II (sFlt-1/ANG-II) ratio could be a good predictor of adverse outcomes, including pneumonia, intensive care unit (ICU) admission, intubation, viral sepsis, and death, among pregnant women with COVID-19. They present work from their own clinical practice and the work of others that confirm the paper’s findings. They also report differences in some of the findings from the Espino-y-Sosa et al. paper and offer possible explanations for the difference. The comparison is presented thoroughly with compelling importance for further research in a larger prospective study of the angiogenic profile during pregnancy in SARs-CoV-2 infected women that are symptomatic versus asymptomatic.
Author Response
We thank the Reviewer for his/her consideration and appreciation of our communication.
Reviewer 2 Report
I found this research letter on the correlation between preeclampsia (PE)and COVID19 very intriguing. The data are clearly presented
and adequately support the conclusions. The fact that pregnant women with
asymptomatic COVID19 are at increased risk of PE due to the pathophysiological
mechanisms described is of considerable clinical interest. The only advice I
suggest to the authors is to add a comment on the fact that the data from this
study give a further reason to push pregnant women to get vaccinated given
the dangers associated with even asymptomatic forms of COVID19.
Author Response
We thank the Reviewer for his/her consideration and appreciation of our communication.
We also thank the Reviewer for the suggestion; we added the comment about the importance of vaccination in pregnancy.